# Identification of distinct phenotypes related to benralizumab responsiveness in patients with severe eosinophilic asthma

Hideyasu Yamada[1,2]*, Masayuki Nakajima[1], Masashi Matsuyama[1], Yuko Morishima[1], Naoki Arai[3], Norihito Hida[2], Taisuke Nakaizumi[2], Hironori Masuko[1], Yohei Yatagai[4], Takefumi Saito[3], Nobuyuki Hizawa[1]

1 Faculty of Medicine, Department of Pulmonary Medicine, University of Tsukuba, Tsukuba, Ibaraki, Japan, 2 Division of Respiratory Medicine, Hitachi Ltd, Hitachinaka General Hospital, Hitachinaka, Ibaraki, Japan, 3 Department of Respiratory Medicine, National Hospital Organization, Ibarakihigashi National Hospital, Tokaimura, Ibaraki, Japan, 4 Division of Respiratory Medicine, Tsukuba Gakuen Hospital, Tsukuba, Ibaraki, Japan

* h.yamada@md.tsukuba.ac.jp

**Data Availability Statement:** All relevant data are within the manuscript.

## Abstract

### Purpose

To characterize the clinical phenotypes of severe eosinophilic asthma based on early responsiveness to benralizumab in terms of forced expiratory volume in 1 second (FEV$_1$) improvement.

### Patients and methods

Sixty-four participants diagnosed with severe eosinophilic asthma and who had completed 4 months of benralizumab treatment were included in this analysis. Pre-treatment clinical factors were compared between responders and non-responders according to improvements in ACT or FEV$_1$. Correlations between the sums of increased Type 2-related inflammatory parameters and changes of ACT or FEV$_1$ were also evaluated before and after the 4-month treatment. A two-step cluster analysis was performed to identify distinct phenotypes related to benralizumab responsiveness in terms of FEV$_1$.

### Results

At the 4-month timepoint, all parameters, except for FeNO, were significantly improved after benralizumab treatment. FEV$_1$ responders were associated with higher levels of Type 2-related inflammatory parameters. An improvement in FEV$_1$ but not in ACT was clearly associated with increases in the sums of increased type 2-related inflammation parameters (p = 0.0001). The cluster analysis identified 5 distinct phenotypes of severe eosinophilic asthma according to the variable FEV$_1$ responsiveness to benralizumab. The greatest response was found in the distinct phenotype of severe eosinophilic asthma, which was characterized by modest increase in total IgE and FeNO relative to blood eosinophils with least exposure to smoking.

**Funding:** The funder "Hitachi Ltd" provided support in the form of salaries for authors [HY, N Hida and TN], but did not have any additional role in the study design, data collection and analysis, decision to publish, or preparation of the manuscript. The specific roles of these authors are articulated in the 'author contributions' section.

**Competing interests:** H.Y. has received lecture fees from AstraZeneca, Sanofi and Boehringer Ingelheim. N. Hizawa is on the GlaxoSmithKline advisory board; has received research support from Astellas, AstraZeneca, Boehringer Ingelheim, GlaxoSmithKline, Kyorin, MSD, Novartis, and Chugai Pharmaceutical; and has received lecture fees from Astellas, AstraZeneca, Boehringer Ingelheim, GlaxoSmithKline, Kyorin, MSD, and Novartis. The rest of the authors declare that they have no relevant conflicts of interest. This does not alter our adherence to PLOS ONE policies on sharing data and materials.

**Abbreviations:** $FEV_1$, forced expiratory volume in 1 second; ACT, Asthma Control Test; FeNO, fractional exhaled nitric oxide; ACO, Asthma-COPD overlap.

## Conclusion

This study, to the best of our knowledge, is the first cluster analysis to report distinct phenotypes related to clinical benralizumab response in a real-world population with severe eosinophilic asthma. These results may help to predict responsiveness to benralizumab in patients with severe eosinophilic asthma.

## Introduction

In recent years, several new biologics, such as benralizumab, have been developed to treat patients suffering from asthma poorly controlled by high-dose ICS and long-acting bronchodilators. Benralizumab is a humanized, interleukin (IL)-5Rα-specific, monoclonal antibody that effectively ameliorates asthmatic episodes by inducing rapid and nearly complete depletion of eosinophils [1,2]. Although it has been reported to significantly improve clinical symptoms in patients with eosinophil counts above 150-300/µl in several international phase III trials, the optimal biologic for treatment of severe asthma varies based on each patient's individual pathophysiology and a key unmet clinical need is a lack of clinically available biomarkers to guide treatment [3]. Traditional double-blind randomized controlled trials often result in a more homogenous patient population regarding demographics and disease characteristics than patients treated in everyday practice and it is necessary to obtain data on real-world outcomes to complement clinical trials and guide treatment-related decisions.

Based on the hypothesis that even severe eosinophilic asthma does not represent a single phenotype of asthma, this multi-center, non-interventional, retrospective observational study in a real-world setting sought to characterize the clinical phenotypes of severe eosinophilic asthma based on early responsiveness to benralizumab in terms of forced expiratory volume in 1 second ($FEV_1$) improvement.

## Material and methods

### Ethical statement

All participants provided written, informed consent and the Ethics Review Committees of the Tsukuba University School of Medicine, the Hitachi Ltd, Hitachinaka General Hospital, and the Ibarakihigashi National Hospital approved the study protocols (IRB number: R01-350).

### Study population

The participants were 64 patients who had been diagnosed with severe eosinophilic asthma and then treated with benralizumab for at least 4 months, the first time-point when therapeutic efficacy is conventionally assessed. All participating patients required treatment with high-dose ICS plus long-acting beta agonists while eighteen patients also required maintenance usage of systemic corticosteroids. All patients had blood eosinophil counts of at least 150 cells/µl for 2 years prior to initiation of benralizumab and started benralizumab therapy between June 2018 and March 2020 at University of Tsukuba University Hospital, National Hospital Organization National Ibarakihigashi Hospital, or Hitachinaka Hospital. We included only patients with data on all of the nine factors (sex, age, $FEV_1$, blood eosinophil counts, total IgE, FeNO, ACT, age of onset and smoking index) used in the cluster analysis.

Spirometry was performed at each hospital in accordance with criteria established by the Japanese Respiratory Society (JRS) [4].

## Statistical analysis

Changes in asthma control test (ACT) scores, $FEV_1$, $\%FEV_1$, fractional exhaled nitric oxide (FeNO), blood eosinophil counts (log transformed), total serum IgE (log transformed), and doses of systemic corticosteroid after 4 months of benralizumab were evaluated by the t-test.

Patients who achieved an ACT score of 25 or a 3 or greater increase in the ACT score at 4 months after starting benralizumab were considered ACT responders (n = 47) while improvement of $FEV_1 \geq 100$ mL at 4 months after treatment was considered as $FEV_1$ responsive (n = 35). Pre-treatment clinical factors were compared by t-test between responders and non-responders in each criterion of 4-month responsiveness to benralizumab. We also used the Jonckheere-Terpstra trend test to evaluate correlations between the sums of increased type 2-related inflammatory parameters (eosinophils, IgE, and FeNO) and any changes of ACT or $FEV_1$ before and after treatment. Cut-off values for these type 2 parameters were as follows: 300 eosinophils/μl, 25 ppb FeNO, and total IgE of 100 IU/ml.

The hypothesis-driven, Two-Step cluster analysis was performed using a set of variables that can be easily collected and used in routine practice; they included changes in $FEV_1$ after 4-month treatment of benralizumab, age, percent-predicted $FEV_1$, pack-years of cigarette smoking, ACT, FeNO, log-transformed peripheral blood eosinophil counts, log-transformed total IgE levels, and age of disease onset. For the cluster analysis, all factors were standardized and the number of clusters that gave the maximum difference in the $FEV_1$ change were defined. The Two-Step Cluster Analysis procedure was conducted with IBM SPSS Statistics, version 24, which uses a likelihood distance measure that assumes independent variables in the cluster model (https://www.spss.ch/upload/1122644952_The%20SPSS%20TwoStep%20Cluster%20Component.pdf).

## Results

Patient characteristics at baseline before initiating benralizumab are shown in Table 1 while Table 2 shows changes in the clinical parameters of 64 patients after 4 months of benralizumab treatment. Overall, all parameters, except for FeNO, were significantly improved after benralizumab treatment; however, we also observed a large variation in the magnitude of treatment response in patients with severe eosinophilic asthma in terms of $FEV_1$ or ACT (Fig 1). In addition, we found no significant correlation between improvements in ACT and changes in $FEV_1$ at the 4-month treatment timepoint (r = 0.2, p = 0.11, Fig 1).

Pre-treatment phenotypic differences between responders and non-responders were examined for each responsiveness criterion ($FEV_1$ and ACT responders) (Table 3). $FEV_1$-responders were associated with higher levels of type 2 inflammation-related parameters; however, no clinical factors, except for baseline ACT, were associated with ACT responders. The number of patients on maintenance oral corticosteroid tended to be higher in $FEV_1$ responders (p<0.1). Correlations of baseline type 2 parameters, including blood eosinophil counts, FeNO, and total IgE levels, with changes in $FEV_1$ or ACT are also shown in Table 4. Baseline levels of FeNO and total IgE were significantly correlated with 4-month changes in $FEV_1$ while no baseline type 2 parameters were correlated with 4-month changes in ACT. When we also associated the sums of increased type 2 inflammation-related parameters with changes in ACT or $FEV_1$ (Figs 2 and 3, Tables 5 and 6), improvement in $FEV_1$, but not in ACT, was strongly associated (p<0.001).

We identified 5 distinct phenotypes of severe eosinophilic asthma in terms of $FEV_1$ response to benralizumab. In addition to the $FEV_1$ changes at 4 months, type 2 inflammation-related parameters differed significantly among these 5 clusters (Fig 4, Table 7).

Cluster D (n = 14), with the greatest improvement in $FEV_1$, was characterized by middle age disease onset, lower smoking exposure, moderate airflow obstruction, and moderately

increased levels of type 2 parameters. Two clusters, B (n = 13) and C (n = 9), showed moderate $FEV_1$ improvements. Cluster B was characterized by the prominent intensity of type 2 inflammation with the lowest lung function and lower smoking exposure while Cluster C was characterized by the mildest symptoms, milder airflow obstruction, high intensity of smoking exposure, and the presence of allergic rhinitis with heightened type 2 parameters. Two clusters, A (n = 20) and E (n = 8), showed no $FEV_1$ improvements over 4 months. Cluster A was characterized by the oldest age at onset, the highest exposures to smoking with impaired lung function and mildly elevated type 2 parameters, corresponding to asthma-COPD overlap. Cluster E was characterized by male dominance, the youngest onset, lower smoking exposure, the mildest impairment of lung function and the highest BMI. It was evident that responsive clusters D, B and C were characterized by enhanced type 2 immunity, as assessed by blood eosinophil counts, FeNO, or total IgE, in comparison to non-responsive clusters A and E (Fig 5, Table 7).

## Discussion

Benralizumab exerts a very rapid and effective therapeutic action in patients with severe eosinophilic asthma [5]. Overall, our study supports the contention that, following 4 months of treatment, type 2-related inflammatory parameters, including peripheral blood eosinophil counts, FeNO, and total serum IgE levels, were associated with $FEV_1$-responsiveness to benralizumab. However, a large variation in the magnitude of the responsiveness to benralizumab was also observed. Most importantly, our cluster analysis of the $FEV_1$ responders clearly indicated the presence of several distinct phenotypes with variable early responsiveness to benralizumab. The greatest response was found in the distinct phenotype of severe eosinophilic asthma (Cluster D), which was characterized by modest increases in total IgE and FeNO relative to blood eosinophils with less smoking exposure. Moderate responses were in Clusters B and C, and poor responses were in Clusters A and E. Interestingly, Cluster B, with the highest levels of type 2 parameters, did not demonstrate the strongest effect of benralizumab. It is likely that the intensity of type 2 inflammation in Cluster B may have overwhelmed the ability of benralizumab to fully suppress it over 4 months and thus Cluster D, with milder type 2 inflammation intensity, paradoxically showed the greatest responsiveness. In addition, since increased levels of IgE and/or FeNO reflect the activity of IL-4 and/or IL-13, but not that of IL-5, benralizumab may have failed to effectively suppress activities of these cytokines which led to the notably milder responsiveness observed in Clusters B and C. Indeed, benralizumab failed to lower the levels of FeNO in the overall patient population. This may also explain why Cluster D, which was characterized by lower levels of total IgE and FeNO in comparison to Cluster B, showed the greatest responsiveness to benralizumab. Elevated levels of total IgE and/or FeNO relative to blood eosinophils may therefore be predictive for reduced responsiveness to benralizumab.

Given that two responsive clusters (B and D) had a low smoking exposure, higher exposures to cigarette smoking, which characterized Clusters A and C, seem to reduce the effect of benralizumab. In fact, smoking exposure has been shown to adversely impact the effectiveness of ICS and accumulating evidence indicates that smoking exposure may induce significant pathobiological changes through predominance of activated macrophages and neutrophils [6]. Persistent exposure to cigarette smoke may therefore drive additive or synergistic inflammatory and remodeling responses in the asthmatic airways, leading to the development of ACO and explaining the impaired responsiveness to benralizumab. In contrast, the poor response seen in Cluster E may be explained by its highest observed BMI levels and possible systemic inflammation from increased levels of several obesity-associated cytokines (TNF-α, IL-6, and

leptin). The presence of non-type 2 inflammation complicated by smoking or obesity might hinder the effect of benralizumab, which primarily targets enhanced type 2 immunity. In fact, we recently reported that, as a result of comprehensive whole blood gene expression analysis focusing on responsiveness to benralizumab, enhanced gene expression related to neutrophilic activity resulted in distinct patient clusters in cases of poor responders with severe eosinophilic asthma [7].

Because the importance of peripheral blood eosinophil count as a biomarker of benralizumab responsiveness has been consistently described [8], physicians in real-world settings may possibly decide to administer benralizumab solely based on peripheral blood eosinophil counts. In this sense, it is important to note that this study clearly indicates that defining detailed phenotypes of severe eosinophilic asthma by considering levels of total IgE and FeNO, as well as the presence of smoking or obesity, could help to guide more appropriate and individualized treatments.

Pooled data analysis from the Phase III studies (SIROCCO and CALIMA) identified oral corticosteroid (OCS) use, nasal polyposis, pre-bronchodilator forced vital capacity (FVC), prior year exacerbations, and age at diagnosis as baseline factors that influenced benralizumab efficacy [9]. In addition, the real-world effectiveness study of benralizumab also highlighted higher eosinophil counts, nasal polyposis, and adult-onset asthma as baseline characteristics associated with a superior response to benralizumab [10]. In the current study, in addition to elevated type-2 markers, lower $FEV_1$ or OCS use appeared to be related to increased $FEV_1$ response to benralizumab, which is in line with the previous studies. In contrast, age at disease onset was not related to the responsiveness in our study, which may be attributed to the rather older disease onset (57.8 years old) observed in our cohort.

In this study, there was no clear correlation between improvements in ACT and $FEV_1$. Given that improvements in ACT were not associated with any of the baseline levels or changes in type 2-related markers, including eosinophils, FeNO, and total IgE, we assumed that ACT-responders were likely to have confounded our results by a placebo effect that relates to the complexity of psychological and biological mechanisms [11]. The disparity between patient perception and actual improvement of type 2 inflammation or airflow obstruction emphasizes the importance of referring to these objective parameters to evaluate clinical responses to benralizumab.

Our study population may not be typical of those often recruited to phase 3 studies of benralizumab. Our participants were older and had a higher proportion of smoking histories, especially in cluster A. Given that this cluster had a rather high number of ACT responders, the effectiveness of benralizumab based on the $FEV_1$ improvement may be underestimated because of the possible coexistence of COPD. The main limitations of the study, therefore, include the small sample size, uncertainty on the generalizability of the findings and whether the assessment of response using changes in $FEV_1$ at 4 months was predictive of the response to other clinical outcomes, such as exacerbation rate at one year.

## Conclusion

This study, to the best of our knowledge, is the first cluster analysis to report distinct clinical phenotypes related to benralizumab response in a real-world population with severe eosinophilic asthma.

Although the results of this study supported the contention that benralizumab was shown to be effective in patients with heightened type 2 inflammatory parameters such as eosinophils, IgE, and FeNO at baseline, our findings also indicated the importance of defining detailed phenotypes of severe eosinophilic asthma, including relative increase in total IgE and/or FeNO to

blood eosinophils or the co-existence of smoking or obesity. Future studies will seek novel molecular pathways underlying these distinct clinical phenotypes. This study may also facilitate larger clustering analyses for the other biologics developed for severe asthma, which will clarify distinctive therapeutic roles by identifying individually responsive phenotypes.

## Acknowledgments

We thank Dr. Bryan J. Mathis of the Medical English Communications Center, University of Tsukuba, for English revision of this manuscript.

## Author Contributions

**Conceptualization:** Hideyasu Yamada.

**Data curation:** Hideyasu Yamada, Masayuki Nakajima, Masashi Matsuyama, Yuko Morishima, Naoki Arai, Norihito Hida, Taisuke Nakaizumi, Hironori Masuko, Yohei Yatagai, Takefumi Saito.

**Formal analysis:** Hideyasu Yamada.

**Investigation:** Hideyasu Yamada.

**Methodology:** Nobuyuki Hizawa.

**Project administration:** Hideyasu Yamada, Nobuyuki Hizawa.

**Supervision:** Nobuyuki Hizawa.

**Validation:** Hideyasu Yamada, Nobuyuki Hizawa.

**Visualization:** Hideyasu Yamada.

**Writing – original draft:** Hideyasu Yamada.

**Writing – review & editing:** Hideyasu Yamada, Nobuyuki Hizawa.

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
