## [Decision Letter · Decision Letter 0]

3 Feb 2021

PONE-D-20-37996

Identification of distinct phenotypes related to benralizumab responsiveness in patients with severe eosinophilic asthma

PLOS ONE

Dear Dr. Yamada,

Thank you for submitting your manuscript to PLOS ONE. We apologise for the delay. After careful consideration, we feel that it has merit but does not fully meet PLOS ONE’s publication criteria as it currently stands. Therefore, we invite you to submit a revised version of the manuscript that addresses the points raised during the review process. Both reviewers raised some concenrs. These concerns are mainly related to low number of patients, to the way that introduction and discussion are presented and fnally to some methodological issues which may affect the interpretation of the results. 

We look forward to receiving your revised manuscript.

Kind regards,

Stelios Loukides

Academic Editor

PLOS ONE

2. Thank you for including your ethics statement:  "All participants provided written, informed consent and the local institutional review board approved the study protocols (IRB number: R01-350)."

3. In your Methods section, please provide additional information about the participant recruitment method and the demographic details of your participants. Please ensure you have provided sufficient details to replicate the analyses such as:

a) the date(s) that you accessed patient data,

b) a description of any inclusion/exclusion criteria that were applied to participant recruitment,

c) a statement as to whether your sample can be considered representative of a larger population, and

d) a description of how participants were recruited.

"H.Y. has received lecture fees from AstraZeneca, Sanofi and Boehringer Ingelheim. N. Hizawa is on the GlaxoSmithKline advisory board; has received research support from Astellas, AstraZeneca, Boehringer Ingelheim, GlaxoSmithKline, Kyorin, MSD, Novartis, and Chugai Pharmaceutical; and has received lecture fees from Astellas, AstraZeneca, Boehringer Ingelheim, GlaxoSmithKline, Kyorin, MSD, and Novartis. The rest of the authors declare that they have no relevant conflicts of interest."

b) We note that one or more of the authors are employed by a commercial company: Hitachi Ltd

(2) Please also provide an updated Competing Interests Statement declaring this commercial affiliation along with any other relevant declarations relating to employment, consultancy, patents, products in development, or marketed products, etc.  

Reviewers' comments:

Reviewer's Responses to Questions

**Comments to the Author**

1. Is the manuscript technically sound, and do the data support the conclusions?

Reviewer #1: Yes

Reviewer #2: Partly

2. Has the statistical analysis been performed appropriately and rigorously? 

Reviewer #1: N/A

Reviewer #2: Yes

3. Have the authors made all data underlying the findings in their manuscript fully available?

Reviewer #1: Yes

Reviewer #2: Yes

4. Is the manuscript presented in an intelligible fashion and written in standard English?

Reviewer #1: Yes

Reviewer #2: Yes

5. Review Comments to the Author

Reviewer #1: Hideyasu Yamada et al, tried to indentify distinct phenotypes related to benralizumab responsiveness in patients with severe eosinophilic asthma on a real world basis study and found 5 phenotypes according to FEV1 responsiveness.

Although it is an interesting study with clinical significance regarding every day clinical practice, there are issues that have to be clarified:

1. The study population section is incomplete as basic characteristics are missing even from the Table 1. It is not clarified if patients have received other kind of biological in the past. What was the number of exacerbations in the previous 1 or 2 years? Besides allergic rhinitis, did they have nasal polyposis as well, and in what percentage?

2. The cut off value of 173 IU/mL for IgE is rather arbitrary and high, as there are many references of 100 IU/mL as normal values. Furthermore, no information is provided about positive skin prick or Rast tests in order to better characterize the phenotype of severe allergic asthma.

3. There is no reference regarding changes in OCS use as almost 18 of the patients were under this treatment. OCS sparing is a major parameter of responsiveness to a biologic agent besides FEV1 improvement. The same applies for the number of exacerbations which should be included in the final analysis.

4. Authors should make a comment comparing their results to those based on pool data of the initial phase III studies of Benralizumab (Bleecker ER, et al. Baseline patient factors impact on the clinical efficacy of benralizumab for severe asthma. Eur Respir J 2018; 52: 1800936) as there are both similarities and differences.

5. The conclusion should be more clear about which phenotypes relate to benralizumab responsiveness.

Minor comments

1. Reference 1 does not support the first paragraph of the introduction section and the same applies for reference 7

2. References 24, 25 at the end of the first paragraph of introduction should be omitted

3. PSL abbreviation in Table 2 is not explained

4. Table 3 is rather confusing and most of the comparisons are not statistically significant, so it is suggested to be omitted. Table 5 could be omitted as Figure 3 is more informative

5. Statistical significant p values lower than 0,05 would better be written as p<0,001 instead of e.g. 3,1 x10-5

Reviewer #2: Thank you for asking me to review this paper.

Cluster analysis was used to investigate benralizumab’ treatment response in a small real-world data set. The study deals with an interest aspect of severe asthma management for a tailored therapy. The paper is well written and easy to follow. My main concerns are the small sample size and the low generalizability of the results.

Comments:

1) Abstract: The paper is about identifying clusters of benralizumab treatment response. However, the Results summary does not clearly define which clusters were identified.

2) Introduction is very generical and not informative. I would emphasize the importance of identifying clusters responding to benralizumab treatment outside clinical trials and to have information from real world settings.

3) Methods: Why not taking into account airway reversibility and post-bronchodilator FEV1? Please specify

4) Results: Table 2 please describe all the abbreviations

5) Discussion needs improvements. Please discuss in detail the published literature in light of the results of your research.

6) Discussion: tables and figures should not be referenced in the Discussion section

7) Discussion: I would suggest adding a paragraph developing the potential usefulness of the study results and their generalizability.

6. PLOS authors have the option to publish the peer review history of their article (what does this mean?). If published, this will include your full peer review and any attached files.

Reviewer #1: No

Reviewer #2: No

---

## [Author Response · Author response to Decision Letter 0]

11 Feb 2021

Responses to the reviewers’ comments

Reviewer: #1

COMMENT 1

The study population section is incomplete as basic characteristics are missing even from the Table 1. It is not clarified if patients have received other kind of biological in the past. What was the number of exacerbations in the previous 1 or 2 years? Besides allergic rhinitis, did they have nasal polyposis as well, and in what percentage?

RESPONSE 1

Thank you for the comment. In the revised manuscript, we have added the number of patients who had previous administration of other biologics at Table 1. We are afraid that information about previous exacerbation and nasal polyps is not available in this real-world cohort.

COMMENT 2

The cut off value of 173 IU/mL for IgE is rather arbitrary and high, as there are many references of 100 IU/mL as normal values. Furthermore, no information is provided about positive skin prick or Rast tests in order to better characterize the phenotype of severe allergic asthma.

RESPONSE 2

Thank you for the comment. According to the suggestion, in the revised manuscript, we have reanalyzed our data using the cut off value of 100 IU/mL, which showed no significant changes in the result. We agree that information about allergen specific IgE responses is important to better characterize each cluster of severe eosinophilic asthma; however, insufficient information about this in this real-world cohort did not allow us to do so.

COMMENT 3

There is no reference regarding changes in OCS use as almost 18 of the patients were under this treatment. OCS sparing is a major parameter of responsiveness to a biologic agent besides FEV1 improvement. The same applies for the number of exacerbations which should be included in the final analysis.

RESPONSE 3

Thank you for the comment. Although this study focused on the initial 4-month after starting benralizumab in a real-world setting, small but significant reduction in doses of maintenance OCS was noted during this period (Table 2). In terms of exacerbation, insufficient information about exacerbation history before starting benralizumab did not allow us to evaluate the effect of benralizumab.

COMMENT 4

Authors should make a comment comparing their results to those based on pool data of the initial phase III studies of Benralizumab (Bleecker ER, et al. Baseline patient factors impact on the clinical efficacy of benralizumab for severe asthma. Eur Respir J 2018; 52: 1800936) as there are both similarities and differences.

RESPONSE 4

Thank you for the comment. We have added a following sentence to the discussion in the revised manuscript.

“Pooled data analysis from the Phase III studies (SIROCCO and CALIMA) identified oral corticosteroid (OCS) use, nasal polyposis, pre-bronchodilator forced vital capacity (FVC), prior year exacerbations and age at diagnosis as baseline factors that influenced benralizumab efficacy (E1). In addition, the real-world effectiveness study of benralizumab also highlighted higher eosinophil counts, nasal polyposis, and adult-onset asthma as baseline characteristics associated with a superior response to benralizumab (E2). In the current study, in addition to elevated type-2 markers, lower FEV1 or OCS use appeared to be related to increased FEV1 response to benralizumab, which is in line with the previous studies. In contrast, age at disease onset was not related to the responsiveness in our study, which may be attributed to the rather older disease onset (57.8 years old) observed in our cohort.” 

E1. Bleecker ER, Wechsler ME, FitzGerald JM, Menzies-Gow A, Wu Y, Hirsch I, et al. Baseline patient factors impact on the clinical efficacy of benralizumab for severe asthma. Eur Respir J 2018 Oct 18;52(4):1800936. doi: 10.1183/13993003.00936-2018.

E2. Joanne EK, Andrew PH, Jaideep D, Gráinne d'A, Abdel D, Cris R, et al. Real-World Effectiveness of Benralizumab in Severe Eosinophilic Asthma. Chest. Online ahead of print. DOI: https://doi.org/10.1016/j.chest.2020.08.2083

COMMENT 5

The conclusion should be more clear about which phenotypes relate to benralizumab responsiveness.

RESPONSE 5

Thank you for the comment. At the 1st paragraph of the discussion, we have now described clearly that the greatest response was found in the distinct phenotype of severe eosinophilic asthma, which was characterized by modest increases in total IgE and FeNO relative to blood eosinophils with least exposure to smoking.

MINOR　COMMENT

1. Reference 1 does not support the first paragraph of the introduction section and the same applies for reference 7

2. References 24, 25 at the end of the first paragraph of introduction should be omitted

3. PSL abbreviation in Table 2 is not explained

4. Table 3 is rather confusing and most of the comparisons are not statistically significant, so it is suggested to be omitted. Table 5 could be omitted as Figure 3 is more informative

5. Statistical significant p values lower than 0,05 would better be written as p<0,001 instead of e.g. 3,1 x10-5

RESPONSE to minor comment

The above has been corrected as appropriate. In the revised manuscript, we omitted Table 5, but not Table 3. We believe that Table 3 has some importance of showing the baseline predictive factors for FEV1 improvement in response to 4-month benralizumab.

Reviewer: #2

COMMENT 1

Abstract: The paper is about identifying clusters of benralizumab treatment response. However, the Results summary does not clearly define which clusters were identified.

RESPONSE 1

Thank you for the comment. In the result section of abstract, we have now described clearly that the greatest response was found in the distinct phenotype of severe eosinophilic asthma, which was characterized by modest increases in total IgE and FeNO relative to blood eosinophils with least exposure to smoking.

COMMENT 2

Introduction is very generical and not informative. I would emphasize the importance of identifying clusters responding to benralizumab treatment outside clinical trials and to have information from real world settings.

RESPONSE 2

Thank you for the comment. We have revised the introduction accordingly.

COMMENT 3

Methods: Why not taking into account airway reversibility and post-bronchodilator FEV1? Please specify

RESPONSE 3

Thank you for the comment. We are afraid that we do not have sufficient information about airway reversibility and post-bronchodilator FEV1 in this cohort.

COMMENT 4

Results: Table 2 please describe all the abbreviations

RESPONSE 4

Thank you for the comment. We have revised Table 2 accordingly.

COMMENT 5

Discussion needs improvements. Please discuss in detail the published literature in light of the results of your research.

RESPONSE 5

We have added a following sentence to the 4th paragraph of the discussion in the revised manuscript.

“Pooled data analysis from the Phase III studies (SIROCCO and CALIMA) identified oral corticosteroid (OCS) use, nasal polyposis, pre-bronchodilator forced vital capacity (FVC), prior year exacerbations and age at diagnosis as baseline factors that influenced benralizumab efficacy (E1). In addition, the real-world effectiveness study of benralizumab also highlighted higher eosinophil counts, nasal polyposis, and adult-onset asthma as baseline characteristics associated with a superior response to benralizumab (E2). In the current study, in addition to elevated type-2 markers, lower FEV1 or OCS use appeared to be related to increased FEV1 response to benralizumab, which is in line with the previous studies. In contrast, age at disease onset was not related to the responsiveness in our study, which may be attributed to the rather older disease onset (57.8 years old) observed in our cohort.”

E1. Bleecker ER, Wechsler ME, FitzGerald JM, Menzies-Gow A, Wu Y, Hirsch I, et al. Baseline patient factors impact on the clinical efficacy of benralizumab for severe asthma. Eur Respir J 2018 Oct 18;52(4):1800936. doi: 10.1183/13993003.00936-2018.

E2. Joanne EK, Andrew PH, Jaideep D, Gráinne d'A, Abdel D, Cris R, et al. Real-World Effectiveness of Benralizumab in Severe Eosinophilic Asthma. Chest. Online ahead of print. DOI: https://doi.org/10.1016/j.chest.2020.08.2083

COMMENT 6

Discussion: tables and figures should not be referenced in the Discussion section

RESPONSE 6

Thank you for your comment. We have revised the manuscript accordingly.

COMMENT 7

Discussion: I would suggest adding a paragraph developing the potential usefulness of the study results and their generalizability.

RESPONSE 7

Thank you for your comment. Accordingly, we have added a sentence as the 3rd paragraph of the Discussion.

---

## [Decision Letter · Decision Letter 1]

24 Feb 2021

Identification of distinct phenotypes related to benralizumab responsiveness in patients with severe eosinophilic asthma

PONE-D-20-37996R1

Dear Dr. Yamada,

We’re pleased to inform you that your manuscript has been judged scientifically suitable for publication and will be formally accepted for publication once it meets all outstanding technical requirements.

Kind regards,

Stelios Loukides

Academic Editor

PLOS ONE

Additional Editor Comments (optional):

Reviewers' comments:

Reviewer's Responses to Questions

**Comments to the Author**

1. If the authors have adequately addressed your comments raised in a previous round of review and you feel that this manuscript is now acceptable for publication, you may indicate that here to bypass the “Comments to the Author” section, enter your conflict of interest statement in the “Confidential to Editor” section, and submit your "Accept" recommendation.

Reviewer #1: All comments have been addressed

Reviewer #2: All comments have been addressed

2. Is the manuscript technically sound, and do the data support the conclusions?

Reviewer #1: Yes

Reviewer #2: Yes

3. Has the statistical analysis been performed appropriately and rigorously? 

Reviewer #1: Yes

Reviewer #2: Yes

4. Have the authors made all data underlying the findings in their manuscript fully available?

Reviewer #1: Yes

Reviewer #2: Yes

5. Is the manuscript presented in an intelligible fashion and written in standard English?

Reviewer #1: Yes

Reviewer #2: Yes

6. Review Comments to the Author

Reviewer #1: Although there are missing data to further supportthe final conclusion, the required questions have been answered and all responses meet formatting specifications

Reviewer #2: The authors addressed all the points raised. The manuscript adds interesting information in the field. I have no further comments.

7. PLOS authors have the option to publish the peer review history of their article (what does this mean?). If published, this will include your full peer review and any attached files.

Reviewer #1: No

Reviewer #2: No

---

## [Editor Report · Acceptance letter]

3 Mar 2021

PONE-D-20-37996R1 

Identification of distinct phenotypes related to benralizumab responsiveness in patients with severe eosinophilic asthma 

Dear Dr. Yamada:

I'm pleased to inform you that your manuscript has been deemed suitable for publication in PLOS ONE. Congratulations! Your manuscript is now with our production department. 

Kind regards, 

on behalf of

Dr. Stelios Loukides 

Academic Editor

PLOS ONE